# Ruthenium Antivenom Inhibits the Defibrinogenating Activity of *Crotalus adamanteus* Venom in Rabbits

**DOI:** 10.3390/ijms25126334

**Published:** 2024-06-07

**Authors:** Vance G. Nielsen

**Affiliations:** Department of Anesthesiology, The University of Arizona College of Medicine, Tucson, AZ 85724, USA; vnielsen@anesth.arizona.edu

**Keywords:** snake venom thrombin-like enzyme, thrombelastography, rabbit, platelet inhibition, ruthenium

## Abstract

Eastern Diamondback Rattlesnake (*Crotalus adamanteus*) envenomation is a medical emergency encountered in the Southeastern United States. The venom contains a snake venom thrombin-like enzyme (SVTLE) that is defibrinogenating, causing coagulopathy without effects on platelets in humans. This investigation utilized thrombelastographic methods to document this coagulopathy kinetically on the molecular level in a rabbit model of envenomation via the analyses of whole blood samples without and with platelet inhibition. Subsequently, the administration of a novel ruthenium compound containing site-directed antivenom abrogated the coagulopathic effects of envenomation in whole blood without platelet inhibition and significantly diminished loss of coagulation in platelet-inhibited samples. This investigation provides coagulation kinetic insights into the molecular interactions and results of SVTLE on fibrinogen-dependent coagulation and confirmation of the efficacy of a ruthenium antivenom. These results serve as a rationale to investigate the coagulopathic effects of other venoms with this model and assess the efficacy of this site-directed antivenom.

## 1. Introduction

Envenomation by a rattlesnake is a medical emergency, and in the Southeastern United States, the Eastern Diamondback Rattlesnake (*Crotalus adamanteus*) inflicts a bite with a uniquely fibrinogenolytic venom that is complex and clinically significant [1,2,3,4,5]. The primary molecular target for this procoagulant, snake venom thrombin-like enzyme (SVTLE)-containing venom, is fibrinogen [1,2,3,4], with the enzymes involved not affecting serine proteases within the coagulation cascade or activating factor XIII [2]. To clarify, the SVTLE is a procoagulant enzyme as it is acting as a selective thrombin, which accelerates clot activation in vitro; however, by removing fibrinogen from the circulation in vivo, the SVTLE is an effective anticoagulant for the envenomed organism. *C. adamanteus* venom contains several snake venom serine proteases (SVSPs) and snake venom metalloproteinases (SVMPs), with one or more of the SVSPs acting as the SVTLE responsible for this proteolytic behavior [4]. Further, the venom appears to have no significant effects on the circulating platelet count or function [1,3,5]. In sharp contrast, envenomation by medically important North American rattlesnakes, such as *Crotalus atrox* (Western diamondback rattlesnake) [6,7], *Crotalus horridus* (timber rattlesnake) [8,9], *Sistrurus miliarius streckeri* (Western pygmy rattlesnake) [10], *Crotalus molossus molossus* (Northern blacktail rattlesnake) [11], and *Crotalus viridis lutosus* (Great Basin rattlesnake) [12], cause complex coagulopathies involving not just fibrinogenolysis but also significant thrombocytopenia. Thus, the proteome of *C. adamanteus* venom provides the opportunity to characterize the effects of the focal molecular loss of fibrinogen on the physical chemistry and coagulation kinetics of thrombus formation.

A methodology capable of documenting the effects of loss of fibrinogen in whole blood or plasma is thrombelastography, which has been utilized in clinical [5] and laboratory investigations [13] involving *C. adamanteus* venom. While one of the first utilizations of thrombelastography/thromboelastometry to assess coagulopathy after envenomation was reported thirty-four years ago [14], the vast majority of clinical and laboratory reports in human and veterinary settings have been published in the last decade, with a few examples cited for the interested reader [15,16,17,18,19,20,21,22,23,24,25,26,27,28,29,30,31,32,33,34]. Facile modifications of thrombelastographic methodologies that allow the assessment of changes in fibrinogen include using either antibodies to block platelet receptor glycoprotein IIb/IIIa to prevent attachment that forms fibrin polymers in whole blood [35], inhibiting the degranulation and formation of the glycoprotein IIb/IIIa by inhibiting microtubular activity with cytochalasin D [34,35,36], or by separating plasma in sodium citrate-anticoagulated whole blood [33]. With the contribution of platelets removed in terms of attachment to fibrin polymers, contraction, and release of factor XIII known to further enhance the speed of clot growth and strength [37], the coagulation kinetics observed involve only the various coagulation pathway proteins, with fibrinogen playing a critical role in final thrombus strength [38]. A depiction of previously described coagulation kinetic variables generated with thrombelastography are displayed in Figure 1.

Given the SVTLE activity identified in *C. adamanteus* venom, it would be predicted that the impact of loss of circulating fibrinogen as a singular molecular target could be detected by discreet changes in thrombelastographic variables. A potential prolongation of TMRTG values could be expected following envenomation, with a loss of MRTG and TTG values in both whole blood and platelet-inhibited samples. Venom-mediated changes would be expected to be greater in magnitude in samples after platelet inhibition. This molecular event and anticipated thrombelastographic data are presented in Figure 2.

The goals of the present study were to firstly characterize the effects of envenomation by *C. adamanteus* venom on coagulation in a recently created sedated rabbit model [32,33], and to secondly determine if the administration of a site-directed ruthenium antivenom [32,33] following envenomation could attenuate the effects of this venom on the coagulation system.

## 2. Results

### Effects of C. adamanteus Venom on Circulating Coagulation Function in Rabbits

Sedated male, New Zealand White rabbits were envenomed in preliminary studies to determine that a dose of 8 mg/kg administered subcutaneously on the right flank would inflict consistent damage to whole blood coagulation as reported with other venoms [32,33]. There was no remarkable change in behavior or distress noted following envenomation, and heart rates and arterial oxygen saturation values were like those observed in the past [32,33]. The two groups of envenomed rabbits consisted of one group that was envenomed (n = 6), and the second group had animals that were envenomed and five minutes later were administered a ruthenium antivenom (n = 5) into the same place as the venom had been injected [32,33]. Whole blood samples were collected from an ear artery catheter prior to envenomation (baseline sample) and then every hour for three hours after envenomation. Blood samples were analyzed with thrombelastography and activated with tissue factor, without or with platelet inhibition as previously described [32,33,34].

As displayed in Figure 3, TMRTG did not significantly increase over time following envenomation in either whole blood (Panel (A)) or whole blood with platelet inhibition (Panel (B)) samples. Consequently, there was no interaction over time between the two groups secondary to antivenom administration.

The MRTG results are depicted in Figure 4. Envenomation resulted in a marked reduction in MRTG values (80–88%) at all three time points following baseline values in whole blood samples without antivenom treatment (Panel (A)). Antivenom administration prevented any significant change in MRTG values in whole blood without platelet inhibition over the three hours following envenomation, and animals administered antivenom had significantly greater MRTG values compared to those not administered antivenom. Lastly, there was a significant interaction of antivenom administration and time.

Data obtained from platelet-inhibited samples (Panel (B)) demonstrated a similar pattern of changes in MRTG values compared to samples without platelet inhibition, but with a few important differences. First, envenomed animals not administered antivenom had a significant decrease in MRTG values compared to baseline values (75–90%); however, despite antivenom administration, a small but significant decrease in MRTG values following envenomation were noted compared to baseline values (24–38%). Nevertheless, animals administered antivenom have significantly greater MRTG values at all three hours following envenomation compared to animals not administered antivenom. Lastly, there was a significant interaction of antivenom administration and time.

Information concerning the effects of envenomation on TTG values is found in Figure 5. TTG values derived from whole blood were significantly decreased (77–81%) compared to baseline values in animals not administered antivenom, whereas animals administered antivenom had no significant decrease in TTG values (Panel (A)). Further, rabbits administered antivenom had TTG values significantly greater than animals without antivenom treatment following envenomation. Lastly, there was a significant interaction of antivenom administration and time.

Regarding data generated from blood samples with platelet inhibition (Panel (B)), again, there was a significant decrease in TTG values compared to baseline values (75–90%) in the group of rabbits envenomed without antivenom treatment. Animals administered antivenom after envenomation demonstrated a significant reduction in TTG values compared to baseline values at the three-hour time point (20%), but not the first-hour and second-hour time points following envenomation. There was a significant interaction of antivenom administration and time. In Figure 6, the %TTG_Plasma_ values at baseline and three hours after envenomation are compared. This value significantly decreased in envenomed animals without antivenom treatment but did not decrease in rabbits administered antivenom. There was a significant interaction of antivenom administration and time.

## 3. Discussion

The present investigation achieved its stated goals. First, thrombelastographic methodology was able to document the exclusive defibrinogenating effects of *C. adamanteus* venom on whole blood and platelet-inhibited coagulation. A significant decrease in MRTG and TTG values in both types of blood sample coupled with a significant decrease in %TTG_Plasma_ strongly supports a discreet molecular event mediated by the SVTLE involving the polymerization of circulating fibrinogen into friable fibrin polymers [1,2]. It was somewhat surprising to see no change in TMRTG values, but it should be noted that platelets serve as the physiological surface for thrombin generation [39], so the initiation of clot formation would be expected to be the last event affected by defibrinogenating whole blood, unlike the prolongation of the onset of coagulation observed in plasma without platelets as fibrinogen concentrations decrease [38]. Put another way, the increased thrombin production observed with the presence of platelet surfaces kinetically will plateau TMRTG values despite having decreased fibrinogen concentrations—which will still be observable as decreases in MRTG and TTG values—given the importance of thrombin activity on TMRTG values as previously demonstrated [38]. Taken as a whole, this animal model provided data consistent with what would be expected from envenomation by *C. adamanteus* as has been reported [1,3,5].

While the utility of thrombelastographic methods in assessing the effects of hemotoxic venom was presented in the Introduction [5,13,14,15,16,17,18,19,20,21,22,23,24,25,26,27,28,29,30,31,32,33,34], the present study serves as an example wherein thrombelastography provides a depth of information not afforded by standard hematological methods. Rather than just documented decreases in fibrinogen concentration with either clotting-based or antigen-based methods following envenomation, the use of thrombelastography without and with platelet inhibition allowed the quantification of the impact of the loss of fibrinogen on all aspects of coagulation (Figure 3, Figure 4, Figure 5 and Figure 6). Further, the effects observed with these methods also take into account the potential influences of fibrin degradation products (FDPs) generated by the SVTLE on coagulation, likely not dissimilar to that observed by FDPs generated by typical thrombolytic processes [40]. The molecular site of action of the SVTLE was confirmed not just by a decrease in the MRTG and TTG values of platelet-inhibited samples but also confirmed by a decrease in %TTG_Plasma_ values. Critically, these thrombelastographic phenomena caused by the SVTLE were markedly decreased or eliminated with ruthenium antivenom. In summary, standard hematological assessments would not provide the kinetic, functional data afforded by thrombelastographic analyses.

The second goal of this work was also achieved as evidenced by the abrogation or attenuation of *C. adamanteus* envenomation-mediated degradation of coagulation of whole blood and platelet-inhibited samples, respectively, by the administration of a ruthenium antivenom. When considering the efficacy of this antivenom in whole blood samples without platelet inhibition (Figure 4 and Figure 5, Panel (A)), the venom-mediated degradation of MRTG and TTG values was prevented by antivenom treatment. However, in samples with platelet inhibition (Figure 4 and Figure 5, Panel (B)), protection from the venom-mediated degradation of fibrinogen was less complete, with a loss of up to 38% of MRTG and 20% loss of TTG values observed. This is not surprising, as it would be anticipated that some venom would either be adsorbed during the 5 min preceding antivenom administration or that the antivenom may not have interacted with all the venom injected. Further, given the diminutive loss of MRTG and TTG values in the treated group, it would be expected that this decrease in MRTG and especially TTG values would be undetectable in the variability of values over time in the whole blood compared to platelet-inhibited samples. Further, ruthenium antivenom treatment prevented a significant decrease in %TTG_Plasma_ values after envenomation with a significant interaction of antivenom administration over time as displayed in Figure 6. The putative mechanism by which ruthenium antivenom exerts efficacy is that ionic compounds formed from RuCl_3_ and the Ru-containing compound tricarbonyldichlororuthenium (II) dimer (CORM-2) in PBS interact with activity-critical histidine residues contained within the snake venom enzymes as seen with the inhibition of *Atheris, Echis,* and *Pseudonaja* venoms by the ruthenium antivenom [41]. In conclusion, ruthenium antivenom significantly inactivated the defibrinating action of the SVTLE in *C. adamanteus* venom as determined by thrombelastographic parameters capable of documenting these molecular events.

This investigation has several unavoidable limitations. For example, there is no way of blinding the author to the group studied as he was the sole experimentalist. The author had to monitor the rabbit, collect and analyze blood samples, envenom the animal, and freshly compound the antivenom described within five minutes of envenomation. Thus, it could be argued that bias could be present. As outlined in the section of this manuscript addressing conflicts of interest, the University of Arizona has oversight of this investigation with the assistance of an uninvolved colleague that can observe experimentation or review primary data. The University also reviews all literature generated by the author post hoc. This process was utilized in previous, recent works [32,33]. Thus, reasonable, institutional mitigation strategies are in place to minimize bias by the author.

In conclusion, the present study was able to utilize a recently constructed rabbit model of envenomation to document the molecular interactions of *C. adamanteus* SVTLE activity with fibrinogen in the circulation with thrombelastographic methods. Further, the efficacy of a ruthenium antivenom in inhibiting this interaction was also documented. These data serve as a rationale to use this same paradigm to characterize the molecular interactions of snake venom toxins on coagulation in vivo and determine the extent of the efficacy of antivenom treatments, either site directed or systemic in administration.

## 4. Materials and Methods

### 4.1. Chemicals and Venoms

Lyophilized venom derived from *C. adamanteus* was provided by the National Natural Toxins Research Center (NNTRC) located at Texas A&M University-Kingsville, Kingsville, TX, USA. The National Institutes of Health fund the NNTRC out of the Office of Research Infrastructure Programs. Venom was dissolved into calcium-free phosphate-buffered saline (PBS, Millipore Sigma, Saint Louis, MO, USA) to a final 30 mg/mL concentration, aliquoted, and maintained at −80 °C. Dimethyl sulfoxide (DMSO), cytochalasin D, tricarbonyldichlororuthenium (II) dimer (CORM-2), and RuCl_3_ were obtained from Millipore Sigma (Saint Louis, MO, USA).

### 4.2. Rabbit Model of Envenomation

Male New Zealand White rabbits (2–3 kg) were procured from Charles River Laboratories (San Diego, CA, USA) and housed within our animal facility and allowed food and water ad libitum for at least 1 week prior to experimentation. The Institutional Animal Care and Utilization Committee of the University of Arizona approved all procedures involving these rabbits (protocol #2022-0887). The protocol was conducted in accordance with all applicable federal and institutional policies, procedures, and regulations. Details concerning rabbit instrumentation, sedation, vital sign monitoring, envenomation, and antivenom administration are available to the interested reader through recent publications [32,33]. Antivenom was composed of CORM-2 10 mg/mL in PBS containing 500 µM RuCl_3_, administered five minutes after envenomation. At the end of experimentation, animals were euthanized with intravenous injection of 1 mL of pentobarbital/phenytoin (390/50 mg/mL).

### 4.3. Coagulation Monitoring

Blood collection and thrombelastographic analyses were performed as recently described in [32,33,34]. Blood samples (1 mL) were collected and immediately placed into a thrombelastographic cup for analysis with a computer-controlled thrombelastograph^®^ haemostasis system (Model 5000; Haemonetics Inc., Braintree, MA, USA) at 39 °C. The mixture used in the series of experiments was composed of 340 µL of whole blood, 10 µL of tissue factor (0.1% final concentration of Pacific Hemostasis™ Prothrombin Time Reagent, Thermo Fisher Scientific, Pittsburgh, PA, USA), and 10 µL of PBS for samples without platelet inhibition or 10 µL of cytochalasin D (5 µM final concentration) in platelet-inhibited samples. After mixing the samples, the following parameters were determined: time to the maximum rate of thrombus generation (TMRTG), this is the time interval (minutes or seconds) observed prior to the maximum speed of clot growth; the maximum rate of thrombus generation (MRTG), this is the maximum velocity of clot growth observed (dynes/cm^2^/second); and total thrombus generation (TTG, dynes/cm^2^), the final viscoelastic resistance observed after clot formation. The %TTG_Plasma_ was calculated by dividing the TTG value of a platelet-inhibited sample by the corresponding TTG value of the whole blood sample without platelet inhibition as previously described [33,34]. Data were collected for 30 min.

### 4.4. Statistical Analyses

Data are presented as mean + SD. All experimental groups were represented by n = 5–6 different animals, as this provides a statistical power >0.8 with *p* < 0.05 using this methodology to assess differences in thrombelastographic parameters within and between groups [32,33]. A commercially available statistical program was used for two-way, repeated measures analyses of variance (ANOVA) as appropriate to the dataset, followed by Holm–Sidak post hoc analyses (SigmaStat 3.1; Systat Software, Inc., San Jose, CA, USA). Graphics were generated with commercially available programs; Origen 2023, OrigenLab Corporation, Northampton, MA, USA; and CorelDRAW 2023, Alludo, Ottawa, ON, Canada). *p* < 0.05 was considered significant.

## Figures and Tables

**Figure 1 ijms-25-06334-f001:**
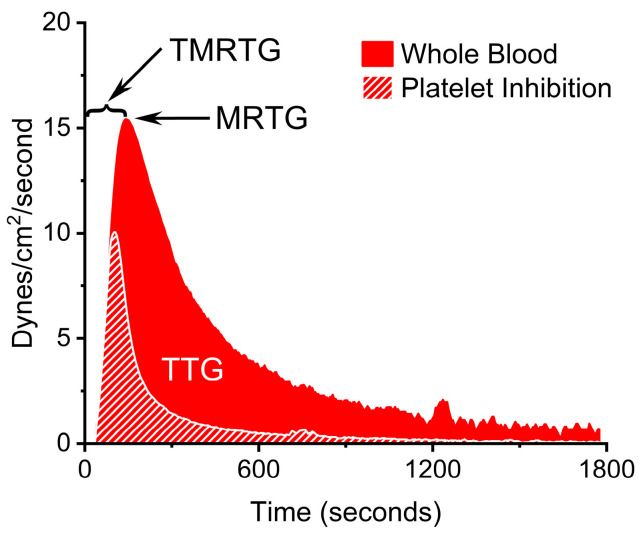
Thrombelastographic variables used in this study typically generated from whole blood (red trace) with the effects of platelet inhibition (white hatched trace) superimposed. TMRTG is defined as time to the maximum rate of thrombus generation (seconds or minutes); MRTG is defined as the maximum rate of thrombus generation (dynes/cm^2^/second); and, TTG is defined as total thrombus generation (dynes/cm^2^), the measure of thrombus resistance-based strength.

**Figure 2 ijms-25-06334-f002:**
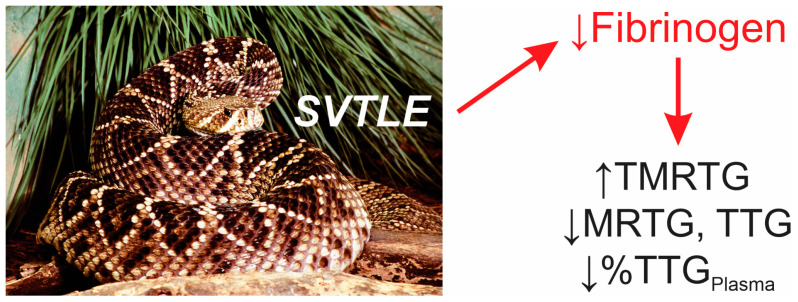
Anticipated experimental outcome. SVTLE from the displayed species, *C. adamanteus*, would compromise coagulation in an envenomed animal by removing fibrinogen from the circulation. Aside from the thrombelastographic variables noted, the loss of clot strength mediated by plasma after platelet inhibition (%TTG_Plasma_) compared to blood without platelet inhibition would be expected to decrease after envenomation. The photograph was kindly provided by the National Natural Toxins Research Center at Texas A&M University-Kingsville, Kingsville, TX, USA.

**Figure 3 ijms-25-06334-f003:**
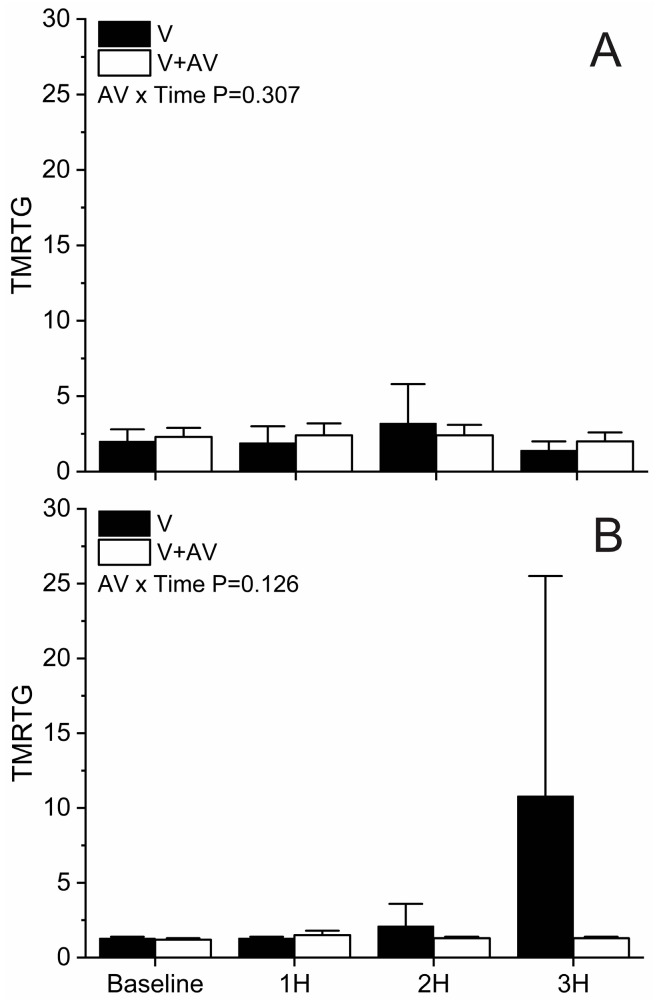
Effects of *C. adamanteus* envenomation on TMRTG. Panel (**A**) represents whole blood results without platelet inhibition, and Panel (**B**) displays results with platelet inhibition. Black bars represent data from animals with envenomation (V), and white bars represent data from envenomed animals administered antivenom (V + AV). Hours after envenomation represented as 1H, 2H, and 3H. Data presented as mean + SD. Two-way analysis of variance (ANOVA) with repeated measures and Holm–Sidak post hoc test were utilized. AV x Time results report interactions of antivenom over time.

**Figure 4 ijms-25-06334-f004:**
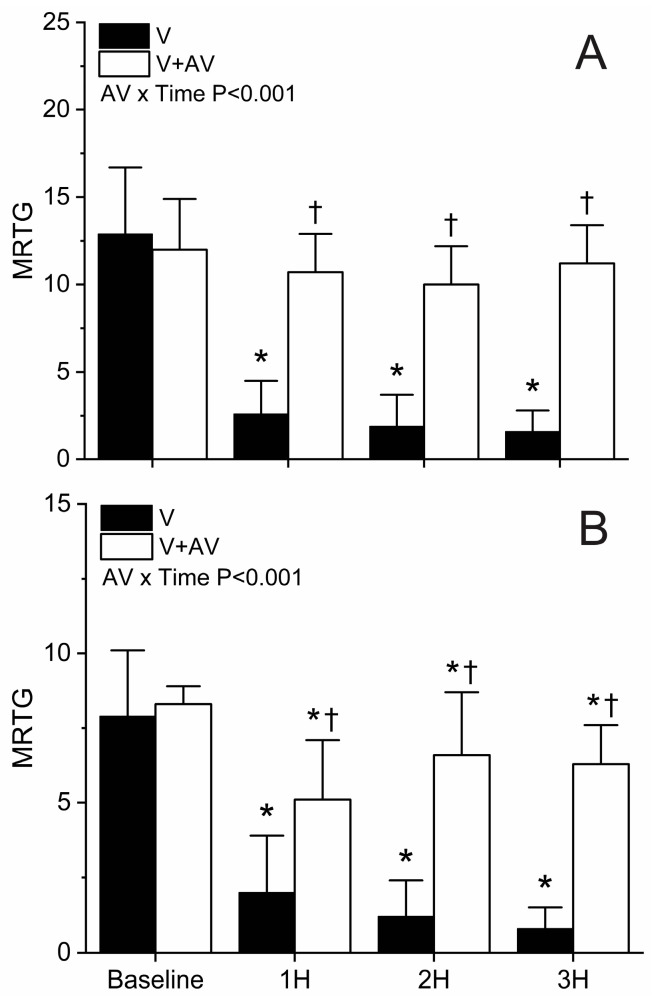
Effects of *C. adamanteus* envenomation on MRTG. Panel (**A**) represents whole blood results, and Panel (**B**) displays results with platelet inhibition. Black bars represent data from animals with envenomation (V), and white bars represent data from envenomed animals administered antivenom (V + AV). Data presented as mean + SD. AV x Time results report interactions of antivenom over time. * *p* < 0.05 vs. Baseline within group; † *p* < 0.05 V vs. V + AV at the same time point.

**Figure 5 ijms-25-06334-f005:**
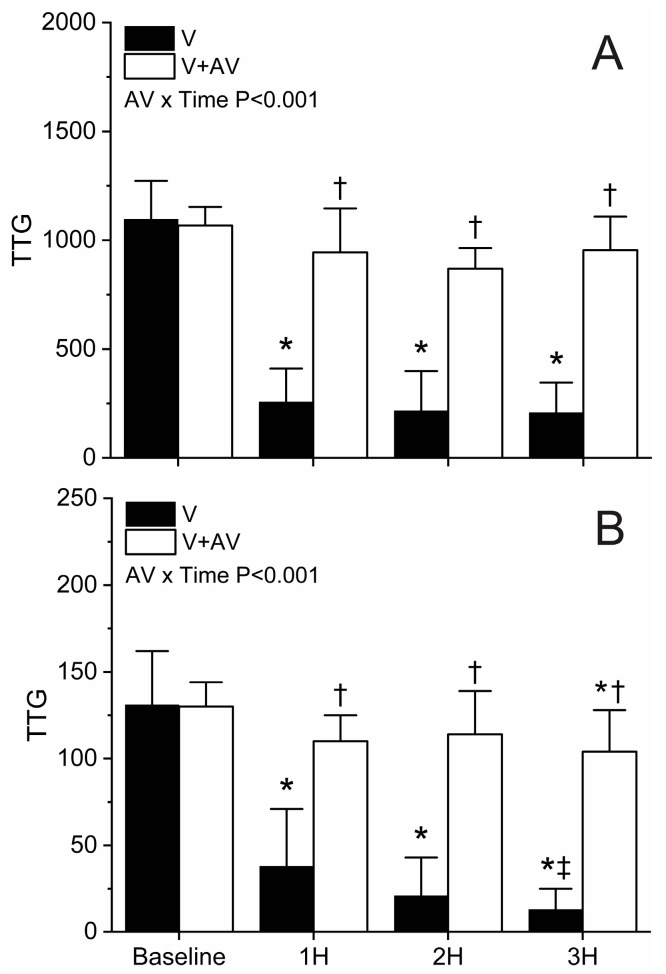
Effects of *C. adamanteus* envenomation on TTG. Panel (**A**) represents whole blood results, and Panel (**B**) displays results with platelet inhibition. Black bars represent data from animals with envenomation (V), and white bars represent data from envenomed animals administered antivenom (V + AV). Data presented as mean + SD. AV x Time results report interactions of antivenom over time. * *p* < 0.05 vs. Baseline within group; † *p* < 0.05 V vs. V + AV at the same time point; ‡ *p* < 0.05 3H vs. 1H within the same group.

**Figure 6 ijms-25-06334-f006:**
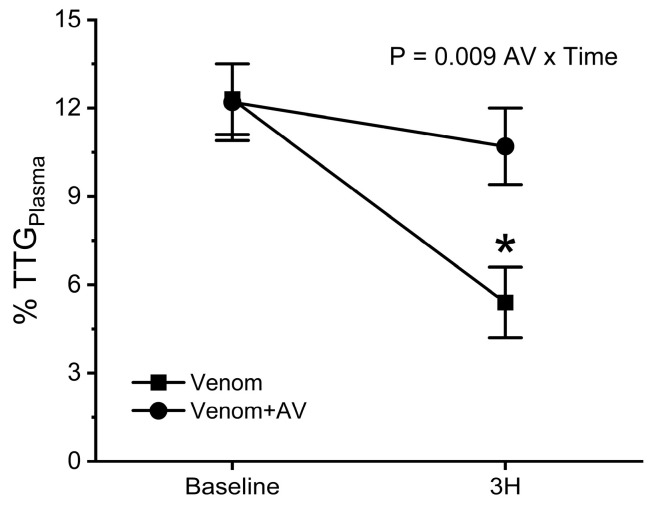
Effects of *C. adamanteus* envenomation on %TTG_Plasma_. Data presented as mean + SD, * *p* < 0.05 vs. baseline. AV x Time results report interactions of antivenom over time.

## Data Availability

Data generated in the conduct of these experiments are presented in the manuscript.

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
