# Peer review of "Ruthenium Antivenom Inhibits the Defibrinogenating Activity of Crotalus adamanteus Venom in Rabbits"

_ijms, 2024, doi:10.3390/ijms25126334_

Round 1

Reviewer 1 Report

Comments and Suggestions for Authors

The manuscript has demonstrated the usefulness of thromboelastographic methods to document coagulopathy kinetically on the molecular level in a rabbit model of snake venom envenomation via analyses of whole blood samples without and with platelet inhibition. However, I think, it can be further improved for the publication.

1. (Ln 10) Spelling check; 'thrombelastographic methods' --> 'thromboelastographic methods'

2. The authors have extensively used thromboelastographic methods as a major tool for the present study. However, there is almost no description about this technique in this manuscript, regarding its necessity for the present work, as well as the strengths and the weaknesses which might be helpful for the people reading this paper. Therefore, the authors shall describe them in somewhere (either Introduction, Materials and methods, or Discussion) of the manuscript.

3.  The authors have studied the therapeutic efficacy of ruthenium antivenom and their mechanism of action for the current work. However, it has not been appropriately described elsewhere within this manuscript for the ruthenium antivenom. In the same context as number 2 above, the authors may address the importance of their ruthenium antivenom.

Author Response

“The manuscript has demonstrated the usefulness of thromboelastographic methods to document coagulopathy kinetically on the molecular level in a rabbit model of snake venom envenomation via analyses of whole blood samples without and with platelet inhibition. However, I think, it can be further improved for the publication.”

            I appreciate the reviewer’s comment and will try to satisfy the request.

“1. (Ln 10) Spelling check; 'thrombelastographic methods' --> 'thromboelastographic methods'”

            The spelling I present has been so for over twenty years. This was the original spelling used, and the introduction of the “o” began when the competing company that manufacture the ROTEM wanted to distinguish itself and separate its publications. I was unaware of this issue until the owner of the TEG producing company brought it to my attention. This explains why some of my original works that are around 20+ years old have the “o” and some do not. The original description of the technique was by the German author, H. Hartert:

HARTERT H.  Thrombelastography, a method for physical analysis of blood coagulation. Z Gesamte Exp Med. 1951;117(2):189-203.

Therefore, out of deference to history and the literature I and other authors have generated, I prefer to use the original term. I ask the indulgence of the reviewer in this matter.

“2. The authors have extensively used thromboelastographic methods as a major tool for the present study. However, there is almost no description about this technique in this manuscript, regarding its necessity for the present work, as well as the strengths and the weaknesses which might be helpful for the people reading this paper. Therefore, the authors shall describe them in somewhere (either Introduction, Materials and methods, or Discussion) of the manuscript.”

            Thank you for this comment. In the second paragraph of the Introduction, I mentioned the history of using thrombelastography for diagnosis and assessment of therapy on the coagulopathies resulting from hemotoxic snake bite with references [5,13-34]. I also provided figure 1, which specifically identified the parameters to be assessed with thrombelastography with real-world data. Lastly, in figure 2, I linked the molecular target (fibrinogen) to the venom enzyme of interest and then identified the anticipated experimental results and changes in the thrombelastographic parameters presented in figure 1. With this presentation in Introduction, I thought that I justified the use of the methods to conduct the present investigation based on all the previous literature cited.  Nevertheless, I am happy to provide a paragraph in the Discussion section of the revised manuscript to again address the necessity for the use of thrombelastography for this sort of investigation. If I have missed a specific issue that the reviewer needs addressed, I would respectfully ask for additional details to assist me in this response.

“3.  The authors have studied the therapeutic efficacy of ruthenium antivenom and their mechanism of action for the current work. However, it has not been appropriately described elsewhere within this manuscript for the ruthenium antivenom. In the same context as number 2 above, the authors may address the importance of their ruthenium antivenom.”

            I appreciate the comment and opportunity to expand on the mechanism of action of the ruthenium antivenom. A new sentence is included in the Discussion section.

Reviewer 2 Report

Comments and Suggestions for Authors

The author used a thrombelastographic method to characterize the fibrinogenolytic effect of Eastern diamondback rattlesnake (Crotalus adamanteus) envenomation on coagulation, and demonstrate the efficacy of a ruthenium antivenom in preventing fibrinogenolysis following the envenomation. The study is interesting since it provides a paradigm to characterize the effect of snake venom toxins on coagulation in vivo and measure the efficacy of antivenom treatments. I only have some minor questions for the author.

1. In line 9. The author says that the venom of C. adamanteus is defibrinogenating which causes coagulopathy. I assume this means the venom is anticoagulant. However, in line 26, the venom is described as procoagulant. Here is confusing.

2. The rationale behind the explanation for not seeing changes in TMRTG values in line 166-171 is hard to follow, especially for non-expert. It is better to explain it more in detail.

3. Was the author blinded for group allocation during data collection and analysis? If there was no blinding, is it possible that the results have been biased?

Author Response

“The author used a thrombelastographic method to characterize the fibrinogenolytic effect of Eastern diamondback rattlesnake (Crotalus adamanteus) envenomation on coagulation, and demonstrate the efficacy of a ruthenium antivenom in preventing fibrinogenolysis following the envenomation. The study is interesting since it provides a paradigm to characterize the effect of snake venom toxins on coagulation in vivo and measure the efficacy of antivenom treatments. I only have some minor questions for the author.”

            I appreciate the kind comments of the reviewer and hope that I will satisfy the issues raised.

“1. In line 9. The author says that the venom of C. adamanteus is defibrinogenating which causes coagulopathy. I assume this means the venom is anticoagulant. However, in line 26, the venom is described as procoagulant. Here is confusing.”

            I appreciate the reviewer’s comment and apologize for the confusion. I have added a sentence to differentiate the in vitro procoagulant action of the venom from the in vivo anticoagulation experienced by the envenomed organism.

“2. The rationale behind the explanation for not seeing changes in TMRTG values in line 166-171 is hard to follow, especially for non-expert. It is better to explain it more in detail.”

            I thank the reviewer for this comment and have composed what I hope is a more complete explanation.

“3. Was the author blinded for group allocation during data collection and analysis? If there was no blinding, is it possible that the results have been biased?”

            I appreciate this comment. It was impossible for me to be blinded, as by design the envenomed group that had no antivenom administered was not injected with a vehicle control. Further, I have to freshly prepare the antivenom from its constituents in under five minutes. Additionally, I alone am sedating the rabbit, monitoring vital signs, and performing thrombelastographic analyses. There is no way to prevent my knowing what was going on as only I perform these experiments.

            When it comes to bias, I am the inventor of this antivenom. To mitigate conflict of interest, the University of Arizona monitors my activities and I have an uninvolved colleague that may observe experimentation and review data.

            While the manuscript has a section describing the conflict of interest and the mitigation plan, I have now added a small limitation paragraph in Discussion to address this matter.